# The Effect of Multiple Freeze–Thaw Cycles on the Microstructure and Quality of *Trachurus murphyi*

**DOI:** 10.3390/foods10061350

**Published:** 2021-06-11

**Authors:** Chunlin Hu, Jing Xie

**Affiliations:** 1College of Food Science and Technology, Shanghai Ocean University, Shanghai 201306, China; hcl306@163.com; 2National Experimental Teaching Demonstration Center for Food Science and Engineering, Shanghai Ocean University, Shanghai 201306, China; 3Shanghai Engineering Research Center of Aquatic Product Processing and Preservation, Shanghai 201306, China; 4Shanghai Professional Technology Service Platform on Cold Chain Equipment Performance and Energy Saving Evaluation, Shanghai 201306, China; 5Collaborative Innovation Center of Seafood Deep Processing, Ministry of Education, Dalian 116034, China

**Keywords:** *Trachurus murphyi*, microstructure, quality, freezing and thawing

## Abstract

Temperature fluctuation in frozen food storage and distribution is the perpetual and core issue faced by the frozen food industry. Ice recrystallisation induced by temperature fluctuations under cold storage causes microstructural changes in fish products and irreversible damages to cells and tissues, which lower the frozen fish quality in the food chain. This study is intended to explore how repeated freezing–thawing affected the microstructure and quality of *Trachurus murphyi* during its frozen storage. The results showed the consistency between the increase in ice crystal diameter, volume, and porosity in frozen fish and the increase in centrifugal loss (from 22.4% to 25.69%), cooking loss (from 22.32% to 25.19%), conductivity (from 15.28 Ms/cm to 15.70 Ms/cm), TVB-N (from 16.32 mg N/100 g to 19.94 mg N/100 g), K-value (from 3.73% to 7.07%), and amino acid composition. The muscle structure change observed by Fourier-Transform Infrared spectroscopy (FT-IR) showed that the content of α-helix reduced from 59.05% to 51.83%, while the β-sheet fraction grew from 15.44% to 17.11%, β-turns increased from 5.45% to 7.58%, and random coil from 20.06% to 23.49%. Moreover, muscular structure exhibited varying degrees of deterioration with increasing cycles of freezing and thawing as shown by scanning electron microscopy (SEM). We studied the muscular morphology, which included the measurement of porosities (%) of pore that increased (from 1.4% to 4.3%) and pore distribution, by X-ray computed tomography (uCT). The cycles of the freeze–thaw resulted in structural changes, which seemed to be closely associated with ultimate quality of frozen fish products.

## 1. Introduction

Temperature fluctuation during frozen food storage and distribution is one of the core issues faced by the frozen food industry. Fluctuation may result in ice recrystallisation and microstructural changes in frozen food products. It possibly causes irreversible damage to cells and tissues and lowers frozen food quality in the logistics, where the threat of seafood commercial frauds is current and growing [1,2].

The phenomenon of recrystallisation can be divided into five types, namely accretive, iso-mass, migratory, irruptive, as well as pressure-induced types [3]. Among these types, the most common recrystallisation types of frozen food are iso-mass, migratory, and accretive. Recrystallisation greatly reduces frozen food product quality when food is deposited at a cold chain terminal, shortens the food’s shelf life, and possibly leads to financial losses to the manufacturer [4]. Hence, preservation of the quality of frozen food for a long time has become a major challenge for this industry.

The rate of the freezing determines the distribution of the ice crystals as well as morphology during their formation. Studies have been carried out for observing microstructural changes of frozen food through a combination of image detection technologies such as microphotography in freeze-dried sections, scanning electron microscopy (SEM) [5], and X-ray computed tomography (CT) [6]. SEM has been applied to ice crystal imaging and quantification of frozen food products. Wang et al. (2018) [7] examined structural changes in muscle fibres of frozen horse mackerel and oval squid with SEM, and investigated how ice crystals potentially affect the flesh quality. X-ray digital imaging has gained wide application as one non-invasive method for assessing the internal structure in biological material. It has been employed for reconstructing the 3D internal structure in apple and peach, and to visualise the density and distribution for video-fluoroscopic tested food material [8]. In addition, CT can be employed for visualising how ice crystals are formed in the freezing process of specific food products [9].

The aim of this study was to analyse ice recrystallisation and evaluate how temperature fluctuations affect the quality of frozen *Trachurus murphyi*. We calculated centrifugal loss and cooking loss rates, conductivity, total volatile basic nitrogen (TVB-N), amino acid composition, free amino acids (FAAs), nutritional value, myofibril protein secondary structure, the thermal performance of muscle proteins, and the microstructure in frozen *T. murphyi* in varying freeze–thaw (F–T) cycles. Moreover, we studied the muscular morphology, which included the measurement of porosities (%) of pore and pore distribution by uCT.

## 2. Materials and Methods

### 2.1. Preparation for F–T Treatment

The F–T treatment procedure involved freezing *T. murphyi* at −18 °C for 48 h and subsequent thawing at 4 °C for 6 h. This procedure needed to be repeated 2, 4, 6, and 8 times to obtain samples under different F–T cycles. Quality parameters related to *T. murphyi* were assessed, and the microstructure was observed after freezing and thawing.

### 2.2. Preparation of Myofibrillar Protein and Sample Treatment

The myofibrillar protein (MP) was extracted using a protocol described previously [10]. Overall, 2 g of thawed and chopped meat was homogenised. Further, resulting homogenate was centrifuged at 10,000 *g*, 4 °C, for 15 min. The supernatant had to be removed. This procedure needed to be repetitively operated twice. The precipitate was rinsed twice with 10-fold volume of 20 mmol/L Tris-maleate and 0.6 mol/L KCL solution under homogenisation. The precipitate was extracted at 4 °C for 3 h and further centrifuged for 15 min at 10,000 rpm; the supernatant contained MP.

### 2.3. Determination of Centrifuge Loss

The water centrifuge loss analysis was performed using a method developed by Zang et al. (2017) [11], with some modifications. Briefly, 2 cm × 1 cm (approximately 2 g) meat portions were sliced from the dorsal muscle for weighting (W_0_). Further, the samples were covered using a filter paper, placed in a polyethylene tube with a round bottom of 50 mm diameter, and centrifuged at 10,000 rpm at 4 °C for 10 min. Samples after centrifuge from filter paper were re-weighed (W_1_). Every sample was analysed thrice. Centrifuge loss was calculated using the following formula:(1)$$\mathrm{Centrifuge}\frac{\mathrm{loss}}{\%}=\frac{W_{0}-W_{1}}{W_{0}}\times 100$$

### 2.4. Determination of Cooking Loss

The thawed samples of *T. murphyi* were weighed (W_2_) and placed inside a plastic bag. The samples were cooked in water at 85 °C, and subsequently, the samples were weighted (W_3_) following rapid drying. The cooking loss of the samples was calculated using the following equation:(2)$$\mathrm{Cooking}\;\mathrm{loss}/\%=\frac{W_{2}-W_{3}}{W_{2}}\times 100$$

### 2.5. Conductivity Analysis

Conductivity of the fish frozen at −18 °C was analysed using a Model ECTestr11 m (EUTECH, Ayer Rajah). Measurements were taken directly with the samples in the refrigerator.

### 2.6. Determination of TVB-N

To determine the TVB-N, 25 mL of the filtered sample was inserted into the distillation tube for steam distillation using Kjeldahl apparatus. TVB-N values could be indicated by mg N/100 g *T. murphyi* sample. TMA could be determined using a protocol described by Yu et al. (2018) [12]. TMA-N values were measured using a standard TMA curve and are indicated by mg N/100 g *T. murphyi* sample.

### 2.7. Determination of K-Values

ATP-related compounds such as HxR and Hx were analysed by HPLC. The mobile phase consisted of 0.05 M phosphate buffer solution. The peak was observed at 254 nm. The K-value could be given by Equation (3):(3)$$K\;\mathrm{value}\;(\%)=\frac{\mathrm{HxR}+\mathrm{Hx}}{\mathrm{ATP}+\mathrm{ADP}+\mathrm{AMP}+\mathrm{IMP}+\mathrm{HxR}+\mathrm{Hx}}\times 100$$
where ADP, AMP, ATP, IMP, Hx, and HxR refer to adenosine diphosphate, adenosine monophosphate, adenosine triphosphate, inosine monophosphate hypoxanthine, hypoxanthine, and riboside, separately.

### 2.8. Amino Acid Composition

Hydrolysed amino acids were analysed using high-speed amino acid analyser by using a protocol described previously [13]. The measurement was repeated thrice, and the amount was expressed as g/100 g of crude protein.

### 2.9. FAA Analysis

FAAs were detected using automatic amino acid analyser (Agilent 1100 Series, Palo Alto, CA, USA), following a protocol developed by Yu et al. (2018) [12]. In brief, FAAs in all thawed samples were extracted twice through homogenisation and centrifugation cycles in succession by using 5% trichloroacetic acid solution (*w*/*v*). Gathered supernatants could be diluted to 25 mL by using the identical solution. A total of 1 mL diluted solution was filtered via one 0.22 μm filter membrane for FAA analysis. Automatic FAA quantification was performed using a programme in line with retention time and peak area under FAA criteria.

### 2.10. Fourier-Transform Infrared Spectroscopy (FT-IR)

Myofibril protein secondary structure was investigated with the approach developed in Ma et al. (2020) [14]. Peakfit 4.12 software was taken for extracting data in 1600–1700 cm^−1^ in Gaussian fitting, and variations in myofibril protein secondary structure were analysed.

### 2.11. Differential Scanning Calorimetry Measurement (DSC)

DSC was chosen for assessing protein thermal stability-related changes by using a method by Zhang et al. (2020) [15] with several adjustments. Briefly, 15 mg samples were weighed on standard aluminium pans. These pans had to be heated from 25 °C to 85 °C. One empty pan functioned as the reference. Both enthalpy values (△H) and peak temperatures (T_max_) could be acquired through DSC thermograms.

### 2.12. SEM Observations

For SEM observations, 3 mm × 3 mm × 1 mm dorsal muscle portions of *T. murphyi* samples were vacuum–dried in oven for 7 days. The surface of *T. murphyi* samples was analysed with SEM S-3400N (Hitachi, Tokyo, Japan) at 5 kV voltage.

### 2.13. Determination of Pores Using X-ray CT Scans

X-ray CT scan (uCT, high-resolution X-ray industrial CT, Xradia 520 Versa, Carl Zeiss AG, Oberkochen, Germany) was performed using a method developed by Wang et al. (2020) [16]. The samples were sliced into 2 cm × 2 cm × 1.5 cm sections before putting them inside a test chamber at −18 °C. Following instrumental preheating, the samples were placed above an operating table and regulated for aligning with X-ray source. Overall, 2000 photos were captured within more than 10 min. Reconstruction of samples’ 3D images was completed through a detector. All photos had denoising treatment with CT 3D Pro, followed by 3D image reconstruction. We measured the pore parameters, which included pore size distribution and porosity, by Particle Analyser plugin within BoneJ plugin in ImageJ [17]. The image-based sample porosity was determined as follows:(4)$$\mathrm{Porosity}/\%=\frac{\mathrm{Total}\;\mathrm{volume}\;\mathrm{of}\;\mathrm{pores}}{\mathrm{Volume}\;\mathrm{of}\;\mathrm{the}\;\mathrm{Region}\;\mathrm{of}\;\mathrm{Interest}}\times 100\%$$

### 2.14. Statistical Analysis

Data analysis was performed using one-way ANOVA, and Duncan’s test was performed in SPSS 26.0. Final results are indicated by mean ± SD.

## 3. Results

### 3.1. Determination of Centrifuge Loss

Loss after centrifugation increased by 3.29% after 8 cycles of the freeze and thaw compared with that after 2 cycles. Meanwhile, centrifuge loss value exhibited a continuous rising trend as the water holding capacity (WHC) decreased; as shown in Table 1, the WHC in all groups decreased with the increasing of the cycle. WHC reduction during F–T cycles is possibly caused by an increase in ice crystals and recrystallisation that greatly harm the muscle tissues and MP structure [18,19]. The possible reason is that after multiple F–T cycles, water redistribution and extracellular ice crystal formation cause mechanical injury to muscle cell integrity, thus generating a direct impact on collagen protein and muscle cell’s capacity in binding and entrapping water.

### 3.2. Determination of Cooking Loss

As shown in Table 1, the loss in WHC due to cooking exhibited a trend similar to that of the centrifugal loss, and cooking loss ranged from 22.32% after the 2nd F–T cycle to 25.19% after the 8th F–T cycle. During cooking, fat melting and protein denaturation resulted in the release of chemically bound water, which reduced WHC [20,21]. With more cycles of freezing and thawing, the gap of the cell’s structure became larger and the water retention became worse.

### 3.3. Conductivity Analysis

Ions are formed by the dissolution of amines in water. The formation of ions increases conductivity [22]. As shown in Table 1, there was an increase in the conductivity as the number of F–T cycle increased. Conductivity electrode signal and fish volatile amines content (TVB-N) were contrasted. The conductivity rise slope appeared to be related to the TVB-N rise slope.

### 3.4. TVB-N Analysis

The mix of primary, secondary, and tertiary amines expressed as volatile amines as well as toxic nitrogen compounds can be considered to be TVB-N compounds [23]. Post-mortem TVB-N levels rely on microbial and enzymatic activity levels, resulting in spoilage; thus, they can be used as the indicators for meat freshness as well as food security [24]. The TVB-N value in the 2nd F–T cycle was 16.32 mg N/100 g (Table 1), which increased gradually (*p* < 0.05) with increasing F–T cycles and approached 19.94 mg N/100 g after the 8th F–T cycle. With the value of the TVB-N increased, the ammonia, dimethylamine, trimethylamine, and similar volatile basic nitrogen compounds that are associated with endogenous enzyme activity and spoilage-causing bacteria increased and indicated that the quality of sample decreased. Previously, studies have shown similar results; YingShao reported that after 5 FT cycles, the TVB-N values of the control and the treated sample were almost identical [25].

### 3.5. K-Value Determination

The K-value, derived from ATP autolytic decomposition, functions as another major index of fish freshness assessment [26]. Figure 1 reveals variations in ATP-related compounds in frozen *T. murphyi* samples. The ratio of the sum between Hx and HxR to the sum of ATP-degrading compounds denotes the K-value [27]. Therefore, a high K-value indicates low freshness. The K-value following the 2nd F–T cycle reached 3.73%, which increased with increasing F–T cycles (7.07% following the 8th F–T cycle). With more cycles of freezing and thawing, the freshness of the sample decreased. Previously, a study has shown that with the increase of F–T times, more IMP might be degraded from ATP via an autolytic process, followed by the further degradation by fish and microbial enzymes into HxR and Hx, thus contributing to the increase of K value [28].

### 3.6. Amino Acid Composition

The hydrolysed amino acid composition in *T. murphyi* is shown in Table 2. Altogether, 16 amino acids, with 9 essential amino acids (EAAs) and 8 nonessential amino acids (NEAAs) were observed from *T. murphyi*. With freezing and thawing, the total amino acid level in *T. murphyi* after the 4th F–T cycle exhibited a considerably greater value than that in other samples (*p* < 0.05). This result confirmed the variation trend in crude protein that showed remarkably greater content in *T. murphyi* muscle (*p* < 0.05). Amino acids prove to be strongly associated with health, immunity, growth, as well as reproductive ability of humans and animals [29]. According to Table 2 results, Glu is considered the richest amino acid of *T. murphyi*, followed by aspartic acid (Asp), lysine (Lys), leucine (Leu), alanine (Ala), as well as arginine (Arg). Glutamate joins peptide and fatty acid synthesis and can regulate the ammonia level in the body. Lysine, an essential amino acid, can promote body development, enhancing immune functions and improving central nervous system functions [30]. Leu regulates the rapamycin signalling pathway and promotes protein synthesis. Arg regulates nutrient metabolism and enhances body immune capacity. The content of EAAs in muscle production is a major nutrition index [31]. The amount of all amino acids except Gly and Ala significantly increased from the 2nd F–T cycle to the 4th F–T cycle and decreased from the 4th F–T cycle to the 8th F–T cycle (*p* < 0.05). The reason for the increase of the amount of all amino acids from the 2nd F–T to the 4th F–T could be due to transition of one kind of amino acid to another through different reactions such as oxidation and deamination. The changes that occurred during storage may also be attributed to complex chemical reactions, such as protein–protein interaction, protein–fat interaction, and the Maillard reaction [32]. As shown in Table 2, with more cycles of the freezing and thawing, the nutrition of the sample index decreased. The decrease in some amino acid contents may be due to degradation to amines, volatile acids, and other nitrogenous substances, or to the subsequent tendency for a portion of the protein to hydrolyse to the free amino acid [33].

### 3.7. FAA Analysis

FAA contributes greatly to the flavour development of aquatic products; it provides various tastes, such as umami, sweetness, and bitterness. As shown in Table 3, the total FAA content in all amino acids, except Asp, Ser, Gly, Ala, Met, and Arg, initially reduced and then increased gradually during the storage process. Beklevik et al. (2005) [34] discovered a similar tendency from sea bass fillets in the frozen storage. The initial increase of FAA might be due to the breakdown of amino acid and dipeptide caused by proteolytic enzymes [35]. Moreover, the progressive decomposition of biochemical reaction-induced polypeptide was possibly the main cause leading to an increase in the total FAA content in the later phase [35,36]. A total of 15 FAAs were observed in the 2nd F–T cycle, and lysine and histidine were major constituents that accounted for 6.4% and 71.9% of the total FAAs, respectively.

### 3.8. FT-IR Analysis

Signals of amide I region (1600–1700 cm^−1^) in the protein FT-IR spectrum were primarily derived from C-O stretching vibrations, whereas the contribution of C-N stretching and N-H bending vibrations was less. The amide I region which was selected for Fourier deconvolution yielded information on the individual infrared peak as shown in Figure 2A. They are usually considered for detecting protein’s secondary structural components and calculating structural percentages [37]. As shown in Figure 2B, the α-helix content clearly reduced from 59.05% to 51.83% (*p* < 0.05), while the β-sheet fraction grew from 15.44% to 17.11%. As indicated in numerous studies, a reduction in the α-helix fraction leads to the increase of the β-sheet fraction, which is usually caused by protein molecule denaturation and unfolding [38]. The α-helical structure could be primarily stabilised with amino hydrogen (-NH) and hydrogen bonds between carbonyl oxygen (-CO) in polypeptide chain; however, it is destroyed by oxidative stress [39]. The content of β-turns and random coil increased from 5.45% to 7.58% and from 20.06% to 23.49%, respectively, possibly because of stable α-helices and β-sheets’ conversion to unstable turn structures induced by oxidative modifications. The exposure of hydrophobic groups results in the reduction of intramolecular hydrogen bonds subjected to strong hydrophobic effects. With an increase in the β-sheet content, α-helix and turn structures decrease [38].

### 3.9. DSC Analysis

The thermal performance of muscle proteins can be used for measuring quality variations in freezing and frozen storage. In DSC, the maximum transition peak temperature usually functions as the indicator of native conformation, and enthalpy suggests the net content in an ordered secondary structure [39]. Water contained by white or red muscle can be mostly found in myofibrils, particularly the thin and thick filaments between spaces. Therefore, myosin and actin affect the WHC in fish, fish product texture, and fish mince’s functional properties. Two main endothermic transitions were found from all samples. The first peak (T_max_ = 56.08 °C) and the smaller second peak (T_max_ = 71.06 °C) denoted myosin and actin, separately.

As shown in Figure 3, thermal stability was reduced with freezing and thawing in MP samples. The reduction of protein thermal stability was possibly associated with the breakdown in intermolecular hydrogen bonds [40]. Cai et al. (2018) [41] reported the weakening effect of local overheating on hydrogen bonds in actin and myosin, leading to partial protein structure instability. The obvious reduction in the transition temperature suggested the reduction in thermal stability of the muscle protein during F–T cycles, and it was probably associated with subunit dissociation due to denaturation of myosin and actin.

### 3.10. SEM Analysis

Typical SEM microstructures in *T. murphyi* muscle at F–T 2, F–T 4, F–T 6, and F–T 8 are shown in Figure 4. The muscular structure of F–T 2 sample seems intact and smooth, and muscle fibres appear to be in close arrangements. The F–T 2 sample has minor spaces between muscle bundles exhibiting one well-ordered structure. Muscular structure exhibited varying degrees of deterioration with increasing cycles of freezing and thawing. For F–T 8 sample, the muscle structure exhibited the most severe degeneration and turned fuzzy and loose.

In the longitudinal section, the fibres seem obvious because of the formed gaps (Figure 4). Likewise, Z-disc losses could be detected following freezing and thawing. Damage to Z-discs seemed more apparent in *T. murphyi* following freezing and thawing. According to previous studies, the F–T procedure might cause shrinkage as well as drip loss in muscle fibres [21,42,43]. Cross-linking in the myosin heavy chain via disulphide and non-disulphide covalent bonds in refrigeration storage was shown to be conducive to high-molecular-weight polymer as well as aggregate formation. It was possibly related to muscles fibres of *T. murphyi*. In transverse sections, muscle bundles became more separated after 8 F–T cycles (Figure 4). Protein denaturation and endomysium disruption, arising from freezing and thawing, might lead to an incompact structure. Muscle fibres’ loose structure and disruption conformed well to *T. murphyi’s* low shear force value. Damaged structure and denaturation in myofibril proteins due to freezing and thawing might be related to the weak WHC of muscles that was shown above.

### 3.11. uCT Analysis

A new technology based on X-ray micro-CT system was proposed for visualising the 2D and 3D structures of ice crystals that are formed during freezing. This technology can characterise protein product’s ice crystal microstructure following freezing [9,44]. High-resolution X-ray industrial CT could be adopted for scanning frozen muscle samples and studying the precise internal structure composition. Figure 5A shows indicators such as pore amount, porosity, as well as pore volume in fish muscle sample. With increasing cycles of freezing and thawing, the pore amount, porosity, as well as pore volume increased. Meanwhile, as the number of F–T cycles increased, pores became irregular and disorganised.

As shown in Figure 5B, porosity in the four groups was in the order of F–T 2 < F–T 4 < F–T 6 < F–T 8, and the sample porosity was 1.4%, 1.5%, 2.6%, and 4.3%, respectively. The result indicated that with more cycles of freezing and thawing, the ice crystals expand and the pores show existence of ice crystals. The reason for ice crystal growth during storage may be temperature fluctuation and equalisation or water diffusion under homothermal conditions. Specifically, during temperature fluctuations, temperature gradients might lead to minor melting of small ice crystals on the surface and water diffusion into large ice crystals [45]. The structure of the muscle was destroyed during freezing and thawing, and this finding was consistent with that of the microstructure analysis. This might gain verification from the second structural changes in myofibrils of *T. murphyi* samples.

The ice density depends on temperature. The void volume detected following freezing refers to the ice crystal volume at the freezing temperature, and it appeared to be slightly lower than the ice crystal volume under the final freezing temperature [9,46]. Another impact brought by freezing is that if sample temperature is initially lowered to −20 °C, then ice content will gently increase as the number of ice crystals increases. In addition, the freezing process may contribute to shrinkage as per product traits and F–T parameters. Such shrinkage may cause porous microstructural changes from deformation to collapse in whole or in some parts.

## 4. Conclusions

Collectively, multiple F–T cycles induced mechanical damage to muscle tissues as well as deterioration of the quality of *T. murphyi*. There was an increase in the centrifugal loss, cooking loss, conductivity, and TVB-N which was observed particularly after 6 F–T cycles. Observation of the microstructure by using SEM and uCT indicated that ice crystal reformation due to repeated F–T cycles caused texture properties to deteriorate, and this was associated with the muscle microstructural damage. Current research suggested that ice crystals might experience certain accretive effects inside frozen fish in the case of temperature fluctuation. Temperature played an important role in pore formation, which possibly affected the texture. This study confirmed the consistency in the increase in ice crystal diameter, volume, and porosity of frozen fish and the increase in centrifugal loss, cooking loss, conductivity, TVB-N, K-value, amino acid composition, as well as muscle structure damage. The cycles of the freeze–thaw resulted in structural changes, which might be associated with the quality of frozen fish. The study can be applied to verify both the quality of the freezing process and the correct maintenance of the cold chain of frozen products during transport and storage phases.

## Figures and Tables

**Figure 1 foods-10-01350-f001:**
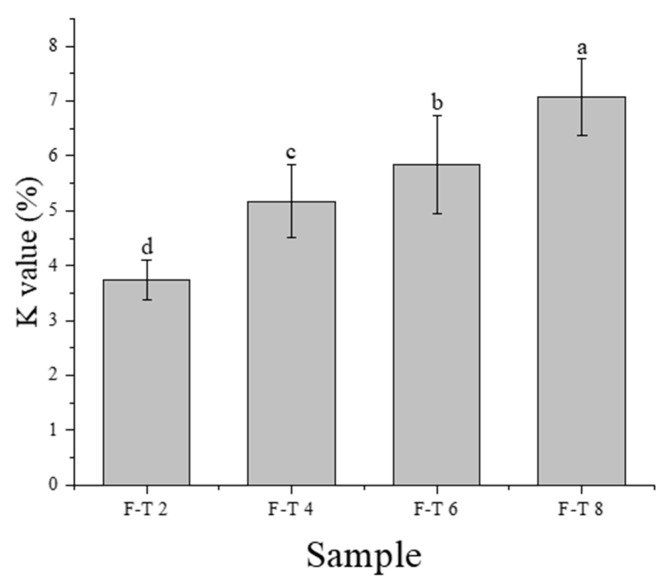
The K-value of the samples in varying freeze–thaw (F–T) cycles.

**Figure 2 foods-10-01350-f002:**
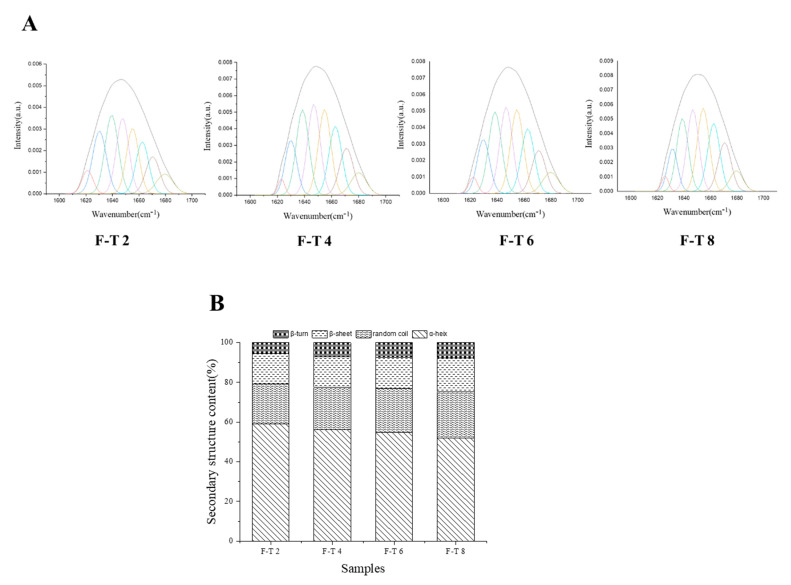
The iterative curve-fitted (**A**) and the proportions (**B**) of the second structure of the sample in varying freeze–thaw (F–T) cycles by FT-IR spectra.

**Figure 3 foods-10-01350-f003:**
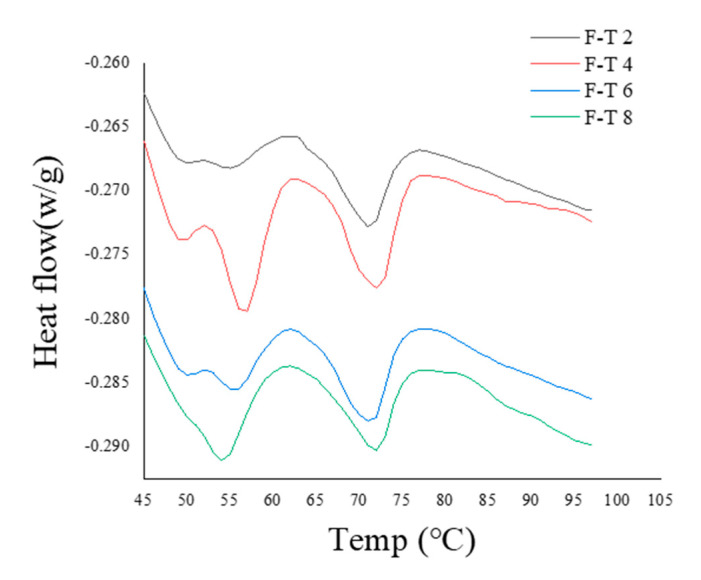
The DSC of the sample with different freezing–thawing (F–T) cycles.

**Figure 4 foods-10-01350-f004:**
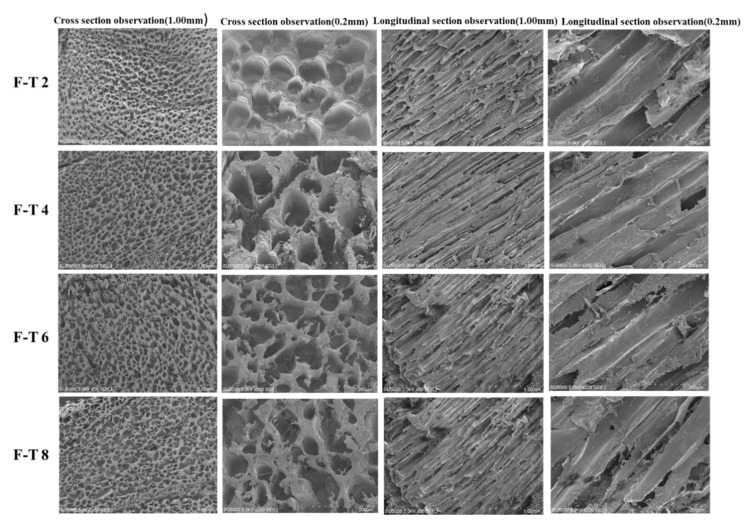
Changes in sample microstructure in various freezing–thawing (F–T) cycles via SEM.

**Figure 5 foods-10-01350-f005:**
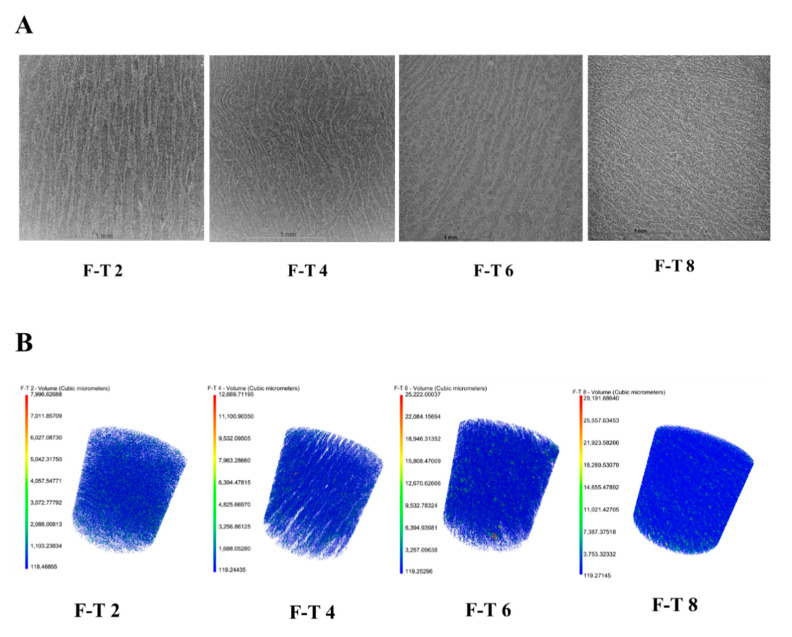
The microstructure (**A**) and the porosities of pore (**B**) of the samples in varying freeze–thaw (F–T) cycles by uCT. Pores are shown in blue colour, resulting in the obvious decreases in total and active sulfhydryl content.

**Table 1 foods-10-01350-t001:** The rate of centrifugal loss and cooking loss, conductivity, and TVB-N of the samples in varying freeze–thaw (F–T) cycles.

Sample	Centrifuge Loss/%	Cooking Loss/%	Conductivity/Ms/cm	TVB-N/mg N/100 g
F–T 2	22.40 ± 1.06 ^d^	22.32 ± 0.51 ^c^	15.28 ± 0.95 ^a^	16.32 ± 0.75 ^d^
F–T 4	23.48 ± 1.02 ^c^	23.46 ± 0.74 ^b^	15.37 ± 0.14 ^a^	18.2 ± 0.15 ^c^
F–T 6	24.33 ± 0.86 ^b^	24.83 ± 1.54 ^b^	15.50 ± 0.07 ^a^	19.47 ± 0.67 ^b^
F–T 8	25.69 ± 0.85 ^a^	25.19 ± 1.00 ^a^	15.70 ± 0.41 ^a^	19.94 ± 0.31 ^a^

^a,b,c,d^ Means in the same column with different superscripts are significantly different (*p* < 0.05).

**Table 2 foods-10-01350-t002:** Changes in the amino acid composition (mg/g) of *T. murphyi* in varying freeze–thaw (F–T) cycles.

Amino Acids	Sample			
	F–T 2	F–T 4	F–T 6	F–T 8
Asp	201.51 ± 5.98 ^d^	214.24 ± 6.42 ^a^	213.50 ± 9.39 ^bc^	184.95 ± 5.19 ^e^
Thr	87.15 ± 2.77 ^c^	94.62 ± 2.85 ^a^	91.58 ± 4.26 ^b^	81.33 ± 2.19 ^d^
Ser	83.94 ± 4.82 ^b^	86.96 ± 2.42 ^a^	85.52 ± 5.06 ^b^	76.75 ± 3.14 ^c^
Glu	280.17 ± 12.49 ^c^	297.81 ± 8.83 ^a^	294.43 ± 15.21 ^b^	253.11 ± 7.73 ^d^
Gly	94.22 ± 2.89 ^c^	94.85 ± 2.89 ^a^	104.64 ± 4.57 ^b^	104.22 ± 2.63 ^d^
Ala	125.32 ± 3.98 ^b^	129.22 ± 3.85 ^a^	133.43 ± 6.05 ^b^	120.87 ± 4.42 ^c^
Cys	11.84 ± 0.41 ^c^	17.57 ± 0.79 ^a^	15.12 ± 0.92 ^b^	10.50 ± 0.62 ^d^
Val	69.69 ± 1.39 ^c^	80.48 ± 2.56 ^a^	72.94 ± 3.13 ^b^	65.59 ± 1.11 ^d^
Met	64.11 ± 1.53 ^c^	69.03 ± 2.18 ^a^	67.72 ± 2.81 ^b^	59.96 ± 1.09 ^d^
Ile	58.79 ± 1.41 ^c^	69.72 ± 2.34 ^a^	61.92 ± 2.54 ^b^	55.83 ± 1.44 ^d^
Leu	150.65 ± 3.94 ^c^	163.91 ± 5.28 ^a^	159.71 ± 6.58 ^b^	140.05 ± 3.28 ^d^
Tyr	70.19 ± 1.32 ^c^	76.97 ± 2.68 ^a^	74.95 ± 2.95 ^b^	65.62 ± 1.26 ^d^
Phe	81.52 ± 1.88 ^c^	86.71 ± 3.17 ^a^	86.22 ± 3.30 ^b^	76.51 ± 1.69 ^d^
Lys	172.55 ± 2.99 ^c^	191.77 ± 6.27 ^a^	185.49 ± 7.57 ^b^	162.19 ± 3.89 ^d^
His	78.88 ± 2.57 ^c^	94.12 ± 3.45 ^a^	92.35 ± 3.35 ^b^	77.01 ± 1.91 ^d^
Arg	110.26 ± 2.51 ^c^	118.08 ± 3.46 ^a^	116.93 ± 5.22 ^b^	106.33 ± 2.29 ^d^

^a,b,c,d,e^ Means in the same column with different superscripts are significantly different (*p* < 0.05).

**Table 3 foods-10-01350-t003:** Changes in the FAA content (ng/100 g) of *T. murphyi* in varying freeze–thaw (F–T) cycles.

FAAs	Sample			
	F–T 2	F–T 4	F–T 6	F–T 8
Asp	10.39 ± 0.25 ^d^	13.73 ± 0.35 ^c^	16.36 ± 0.34 ^b^	30.20 ± 2.07 ^a^
Thr	274.53 ± 10.07 ^e^	176.65 ± 9.92 ^d^	157.24 ± 22.86 ^bc^	296.4 ± 13.71 ^a^
Ser	134.53 ± 4.45 ^d^	149.38 ± 13.82 ^c^	149.9 ± 13.62 ^b^	226.77 ± 3.98 ^a^
Glu	577.70 ± 61.78 ^c^	530.06 ± 21.59 ^d^	612.19 ± 67.42 ^b^	743.68 ± 16.47 ^a^
Gly	201.31 ± 20.43 ^d^	216.13 ± 16.60 ^c^	281.08 ± 9.87 ^b^	292.55 ± 9.73 ^a^
Ala	383.85 ± 19.29 ^d^	446.49 ± 11.54 ^c^	499.73 ± 12.92 ^b^	522.67 ± 34.77 ^a^
Val	336.27 ± 33.93 ^d^	281.10 ± 1.89 ^c^	307.64 ± 8.57 ^b^	425.05 ± 14.86 ^a^
Met	141.40 ± 8.93 ^d^	144.94 ± 7.32 ^c^	171.70 ± 15.05 ^b^	239.38 ± 17.97 ^a^
Ile	95.51 ± 7.68 ^c^	72.11 ± 13.03 ^d^	121.52 ± 4.18 ^b^	151.53 ± 12.07 ^a^
Leu	160.24 ± 8.73 ^c^	140.46 ± 12.52 ^d^	303.34 ± 10.99 ^b^	352.98 ± 7.90 ^a^
Tyr	112.06 ± 8.80 ^c^	106.51 ± 9.92 ^d^	137.39 ± 11.00 ^b^	281.46 ± 11.67 ^a^
Phe	157.91 ± 15.77 ^c^	149.52 ± 14.00 ^d^	719.27 ± 10.87 ^b^	846.54 ± 32.31 ^a^
Lys	647.97 ± 12.98 ^c^	579.72 ± 14.22 ^d^	972.78 ± 34.61 ^b^	1093.79 ± 16.83 ^a^
His	7194.97 ± 103.42 ^c^	6240.25 ± 293.92 ^d^	9199.40 ± 132.47 ^a^	7987.21 ± 225.89 ^b^
Arg	75.36 ± 10.89 ^d^	110.50 ± 13.33 ^c^	137.58 ± 14.46 ^b^	156.79 ± 8.48 ^a^

^a,b,c,d,e^ Means in the same column with different superscripts are significantly different (*p* < 0.05).

## Data Availability

All the data are provided in the article.

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
