# Peer review of "The Effect of Multiple Freeze–Thaw Cycles on the Microstructure and Quality of Trachurus murphyi"

_foods, 2021, doi:10.3390/foods10061350_

Round 1
Reviewer 1 Report
How realistic are freezing and thawing cycles? Can you give any real life examples of this happening and how often?
I am confused about the replication in this experiment and how the standard deviations were determined. Nowhere in the manuscript does it say how many samples were used or even how many fish were tested. This needs to be addressed to legitimize the statistics.
Still need to clarify why amino acids increased between the 2nd and 4th F-t cycle and then decreased from 4th to 8th F-T cycle. Elaborate a bit more
The FAA analysis doesn’t really make sense. Elaborate more.
Conclusions are good but could relate back to why the research is important to the frozen seafood industry.
Line 39 – Should read “divided INTO 5 types”
Line 43 – Incurs is the wrong word the way it is written, consider rewording sentence or picking different word like “leads to”
Line 46 – Suggest changing decides to determines
Line 46 -Should read crystal or crystals’
Line 178 – incomplete sentence or thought “and the juice loss”…what about the juice loss?
Author Response
Dear editors,
Thank you for the reviewers’ useful comments, which give us a big help for the later research. The manuscript has been revised accordingly, and the detailed corrections are listed below point by point:
Reviewer(s)’ Comments to Author:
Reviewer 1:
How realistic are freezing and thawing cycles? Can you give any real life examples of this happening and how often?
Answer: During the tansportion of the frozen fish, there is a cycle of the freezing and thawing if the refrigeration system didn’t work or discharged the frozen fish, another example is that forzen fish was shown on the table and took back into the freezer if not sold out in the supermarket.
I am confused about the replication in this experiment and how the standard deviations were determined. Nowhere in the manuscript does it say how many samples were used or even how many fish were tested. This needs to be addressed to legitimize the statistics.
Answer: Acturely we took 200 fish into the experiment, and more than three fish tested for each item.
Still need to clarify why amino acids increased between the 2nd and 4th F-t cycle and then decreased from 4th to 8th F-T cycle. Elaborate a bit more
Answer: Modified. The amount of all amino acids except Gly and Ala significantly increased from the 2nd F–T cycle to the 4th F–T cycle and decreased from the 4th F–T cycle to the 8th F–T cycle (p < 0.05). The reason of increase of the amount of all amino from the 2nd F-T to 4th F-T could be due to transition of one kind of amino acid to an-other through different reac-tions such as oxidation, deamination, the changes that occurred during storage may also be attributed to complex chem-ical reactions, such as protein-protein interaction, protein-fat interaction and the Maillard reaction [32]. As shown in Table 2, with the more cycles of the freezing and thawing, the nutrition of the sample index decreased. The decrease in some amino acid contents may be due to degradation to amines, volatile acids and other nitrogenous substances, or to the sub-sequent tendency for a portion of the protein to hydrolyze to the free amino acid [32,33].
The FAA analysis doesn’t really make sense. Elaborate more.
Answer: Modified. The initial increase of FAA might be due to the breakdown of amino acid and dipeptide caused by proteolytic enzymes [35]
Conclusions are good but could relate back to why the research is important to the frozen seafood industry.
Answer: Modified. The study can be applied to verify both the quality of the freezing process and the cor-rect maintenance of the cold chain of frozen products during transport and storage phases.
Line 39 – Should read “divided INTO 5 types”
Answer: added the word‘into’
Line 43 – Incurs is the wrong word the way it is written, consider rewording sentence or picking different word like “leads to”
Answer: Modified. Leads to instead of incurs
Line 46 – Suggest changing decides to determines
Answer: Modified. The rate of the freezing determined the distribution of the ice crystals
Line 46 -Should read crystal or crystals’
Answer: Modified. The rate of the freezing determined the distribution of the ice crystals
Line 178 – incomplete sentence or thought “and the juice loss”…what about the juice loss?
Answer: Modified. The juice loss means the juice of the fish meat lost, to avoid mistake, deleted it.
We appreciate your suggestions to the manuscript, they help us to do our research better. The revised manuscript had been resubmitted to the journal. We are looking forward to the positive response.
Yours sincerely
Jing Xie and Chunlin Hu

Reviewer 2 Report
Report of the manuscript - Foods-1248527
The work investigated the effect of multiple freeze-thaw cycles on the microstructure and quality of Trachurus murphyi
The paper is interesting, however, it needs some revisions.
Based on this, I invite authors to resubmit the manuscript after addressing the comments below. Lastly, at the end of scientific revision, English language editing is recommended.
Lines 17-17: please check if is correct to close the sentence with “ during its refrigerated storage”. In fact, according to the U.S. Food and Drug Administration (FDA) refrigerator temperature range between 35° and 38°F (or 1.7 to 3.3°C).
Lines 19-27: the sentence is too long. I suggest breaking it into shorter sentences to help your reader.
Lines 37-38: at a worldwide level, It is recognized that freezing-thawing of seafood products increase their overall spoilage and, when forced, is also considered a “commercial fraud”. Moreover, these practices represent not only a health guarantee for the consumers but also a loss of the economic value of the product. About this concern as well as similar studies on another Trachurus species, please compare your work with the following article where the FT changes have been analysed by mitochondrial enzymes and enzyme extracts from blood cells and lysosomal enzymes): https://www.sciencedirect.com/science/article/pii/S0308814613014453?casa_token=8ssPxMaOhrsAAAAA:wOSnEOFIXbDraHu9WfpntKAnll-XbHsSvsBPY8tQtdfhNBjtMaD91Eydj3wRCkza8OSbBICGow
Lines 70-71_ In the sentence "This procedure had to be repeated for 2, 4, 6 and 8 times to obtain samples with different F – T cycles", It may also be useful to provide the time (days, months ...?) that passed between one FT cycle and the next.
Lines 74-81: if the paragraph “2.2. Preparation of myofibrillar protein and sample treatment” is propaedeutic to “2.10. Fourier-Transform Infrared spectroscopy (FT-IR)”, could be useful putting them one after the other?
Line 99: In the following sentence “Measurements were taken directly with the samples in the refrigerator”, please check if it is correct “refrigerator” or “freezer”.
Lines 130-133: which kind of FTIR spectrometer has been used?
Line 159: “3. Results” or “3. Results and discussion”?
Lines 190-195: Authors should discuss these results, specifically if previous studies on FT cycles on similar seafood products have been published.
Lines 201-203: as before, if previous studies faced FT in similar products, they should be used for an appropriate comparison
Lines 223-227: please check the following sentence “The reason of increase of the amount of all amino from the 2nd F-T to 4th F-T may be the fast freezing efficiently decreased FAA loss, and it also accounted for the peak value of total EAAS that occurred in ice crystals in frozen storage. The content of EAAs in muscle production is a major nutrition index [28]. As shown in Table 2, with the more cycles of the freezing and thawing, the nutrition of the sample index decreased”.
Specifically:
- The authors used “fast freezing”(i.e. -35°C and lower) or, as highlighted in paragraph 2.1. (Preparation for F–T treatment), the sample was simply placed in a standard freezer at -18°C?
- The increase/decrease of AA during the FT cycles could be better explained and discussed.
- Moreover, the final loss of AA, could be due to their decarboxylation and formation of respective biogenic amines?
Fig. 2 A: in the text, where this figure is commented?
Lines 250-251: the values showed in figure 2.b. do not seem to coincide with those shown in the text (39.69% to 35.14% (P < 0.05), while the β-sheet fraction grew from 10.37% to 11.60%).
Author Response
Dear editors,
Thank you for the reviewers’ useful comments, which give us a big help for the later research. The manuscript has been revised accordingly, and the detailed corrections are listed below point by point:
Reviewer(s)’ Comments to Author:
Report of the manuscript - Foods-1248527
The work investigated the effect of multiple freeze-thaw cycles on the microstructure and quality of Trachurus murphyi
The paper is interesting, however, it needs some revisions.
Based on this, I invite authors to resubmit the manuscript after addressing the comments below. Lastly, at the end of scientific revision, English language editing is recommended.
Lines 17-17: please check if is correct to close the sentence with “ during its refrigerated storage”. In fact, according to the U.S. Food and Drug Administration (FDA) refrigerator temperature range between 35° and 38°F (or 1.7 to 3.3°C).
Answer: Modified. Changed the refrigerated storage into frozen storage
Lines 19-27: the sentence is too long. I suggest breaking it into shorter sentences to help your reader.
Answer: Modified. It divided into three sentence structure.
Lines 37-38: at a worldwide level, It is recognized that freezing-thawing of seafood products increase their overall spoilage and, when forced, is also considered a “commercial fraud”. Moreover, these practices represent not only a health guarantee for the consumers but also a loss of the economic value of the product. About this concern as well as similar studies on another Trachurus species, please compare your work with the following article where the FT changes have been analysed by mitochondrial enzymes and enzyme extracts from blood cells and lysosomal enzymes): https://www.sciencedirect.com/science/article/pii/S0308814613014453?casa_token=8ssPxMaOhrsAAAAA:wOSnEOFIXbDraHu9WfpntKAnll-XbHsSvsBPY8tQtdfhNBjtMaD91Eydj3wRCkza8OSbBICGow
Answer: Add the piont. Sincerely it’s a good point, we believe that some test like microstructure and the enzyme tests in correlation with the sensory tests would be useful for the fish industry to unmask fraud in commercial species of blue fish both small and large.
Lines 70-71_ In the sentence "This procedure had to be repeated for 2, 4, 6 and 8 times to obtain samples with different F – T cycles", It may also be useful to provide the time (days, months ...?) that passed between one FT cycle and the next.
Answer: at −18 °C for 48 h and subsequent thawing at 4 °C for 6 h as a cycle.
Lines 74-81: if the paragraph “2.2. Preparation of myofibrillar protein and sample treatment” is propaedeutic to “2.10. Fourier-Transform Infrared spectroscopy (FT-IR)”, could be useful putting them one after the other?
Answer: yes, acturely, we did the experiment one after the other as soon as possible.
Line 99: In the following sentence “Measurements were taken directly with the samples in the refrigerator”, please check if it is correct “refrigerator” or “freezer”.
Answer: Modified. Freezer
Lines 130-133: which kind of FTIR spectrometer has been used?
Answer: Add the model. FT-IR, PerkinElmer, detector DTGS
Line 159: “3. Results” or “3. Results and discussion”?
Answer: Modified. 3. Results and discussion
Lines 190-195: Authors should discuss these results, specifically if previous studies on FT cycles on similar seafood products have been published.
Answer: Modified.Previously studies have shown the similar results, YingShao reported that after 5 times of FT cycles, the TVB-N values of the control and the treated sample were almost identical [24]
Lines 201-203: as before, if previous studies faced FT in similar products, they should be used for an appropriate comparison
Answer: Modified. Previously study shown that with the increase of F-T times, more IMP might be de-graded from ATP via an autolytic process, followed with the further degradation by fish and microbial enzymes into HxR and Hx, thus contributing to the increase of K value [27].
Lines 223-227: please check the following sentence “The reason of increase of the amount of all amino from the 2nd F-T to 4th F-T may be the fast freezing efficiently decreased FAA loss, and it also accounted for the peak value of total EAAS that occurred in ice crystals in frozen storage. The content of EAAs in muscle production is a major nutrition index [28]. As shown in Table 2, with the more cycles of the freezing and thawing, the nutrition of the sample index decreased”.
Answer: Modified.The amount of all amino acids except Gly and Ala significantly increased from the 2nd F–T cycle to the 4th F–T cycle and decreased from the 4th F–T cycle to the 8th F–T cycle (p < 0.05). The reason of increase of the amount of all amino from the 2nd F-T to 4th F-T could be due to transition of one kind of amino acid to an-other through different reac-tions such as oxidation, deamination, the changes that occurred during storage may also be attributed to complex chem-ical reactions, such as protein-protein interaction, protein-fat interaction and the Maillard reaction [32]. As shown in Table 2, with the more cycles of the freezing and thawing, the nutrition of the sample index decreased. The decrease in some amino acid contents may be due to degradation to amines, volatile acids and other nitrogenous substances, or to the sub-sequent tendency for a portion of the protein to hydrolyze to the free amino acid [32,33].
Specifically:
- The authors used “fast freezing”(i.e. -35°C and lower) or, as highlighted in paragraph 2.1. (Preparation for F–T treatment), the sample was simply placed in a standard freezer at -18°C?
Answer: one bag one fish placed in a standard freezer at -18°C for 48h
- The increase/decrease of AA during the FT cycles could be better explained and discussed.
Answer: yes. Modified. The amount of all amino acids except Gly and Ala significantly increased from the 2nd F–T cycle to the 4th F–T cycle and decreased from the 4th F–T cycle to the 8th F–T cycle (p < 0.05). The reason of increase of the amount of all amino from the 2nd F-T to 4th F-T could be due to transition of one kind of amino acid to an-other through different reac-tions such as oxidation, deamination, the changes that occurred during storage may also be attributed to complex chem-ical reactions, such as protein-protein interaction, protein-fat interaction and the Maillard reaction [32]. As shown in Table 2, with the more cycles of the freezing and thawing, the nutrition of the sample index decreased. The decrease in some amino acid contents may be due to degradation to amines, volatile acids and other nitrogenous substances, or to the sub-sequent tendency for a portion of the protein to hydrolyze to the free amino acid [32,33].
- Moreover, the final loss of AA, could be due to their decarboxylation and formation of respective biogenic amines?
Answer: yes. Modified. The decrease in some amino acid contents may be due to degradation to amines, vola-tile acids and other nitrogenous substances, or to the subsequent tendency for a portion of the protein to hydrolyze to the free amino acid. [32,33].
Fig. 2 A: in the text, where this figure is commented?
Answer: The amide I region was selected to Fourier deconvolution yielded information on the individual infrared peak as shown Fig 2A.
Lines 250-251: the values showed in figure 2.b. do not seem to coincide with those shown in the text (39.69% to 35.14% (P < 0.05), while the β-sheet fraction grew from 10.37% to 11.60%).
Answer: thanks, I forgot the normalized to 100%, already corrected.
We appreciate your suggestions to the manuscript, they help us to do our research better. The revised manuscript had been resubmitted to the journal. We are looking forward to the positive response.
Yours sincerely
Jing Xie and Chunlin Hu

Round 2
Reviewer 2 Report
Overall, the manuscript has been sufficiently improved to warrant publication in Foods. Only a minor remark is listed below:
Lines 35-39- Considering that the subject is the “temperature fluctuation”, it does not seem appropriate to state that the consequent tissue damage helps “fish industry to unmask fraud in commercial specie fish”, but it might be more correct to say that “it leads to commercial fraud.
Based on this, please evaluate if the following sentence may better fit in the text:
“Temperature fluctuation during frozen food storage and distribution is one of the core issues faced by the frozen food industry. Fluctuation may result in ice recrystallisation and microstructural changes in frozen food products. It possibly causes irreversible damages to cells and tissues and lowers frozen food quality in the logistics, where the threat of seafood commercial frauds is current and growing [1,2].